# Performance of Reverse Electrodialysis System for Salinity Gradient Energy Generation by Using a Commercial Ion Exchange Membrane Pair with Homogeneous Bulk Structure

**Esra Altıok** [1], **Tuğçe Zeynep Kaya** [1], **Enver Güler** [2,*], **Nalan Kabay** [1,*] and **Marek Bryjak** [3]

1 Department of Chemical Engineering, Faculty of Engineering, Ege University, Izmir 35100, Turkey; altiokesra@gmail.com (E.A.); tugce-zeynep@hotmail.com (T.Z.K.)
2 Department of Chemical Engineering, School of Engineering, Atılım University, Ankara 06830, Turkey
3 Department of Polymer and Carbon Materials, Faculty of Chemistry, Wrocław University of Science and Technology, 50-370 Wrocław, Poland; marek.bryjak@pwr.edu.pl
* Correspondence: enver.guler@atilim.edu.tr.com (E.G.); nalan.kabay@ege.edu.tr (N.K.); Tel.: +90-3125868265 (E.G.); +90-2323112290 (N.K.)

**Abstract:** Salinity gradient energy is a prominent alternative and maintainable energy source, which has considerable potential. Reverse electrodialysis (RED) is one of the most widely studied methods to extract this energy. Despite the considerable progress in research, optimization of RED process is still ongoing. In this study, effects of the number of membrane pairs, ratio of salinity gradient and feed velocity on power generation via the reverse electrodialysis (RED) system were investigated by using Fujifilm cation exchange membrane (CEM Type 2) and FujiFilm anion exchange membrane (AEM Type 2) ion exchange membranes. In the literature, there is no previous study based on a RED system equipped with Fujifilm AEM Type II and CEM Type II membranes that have homogeneous bulk structure. Using 400 μm of intermembrane distance, maximum obtainable power density by 5 pairs of Fujifilm membranes at 1:45 salinity ratio and with a linear flow rate of 0.833 cm/s was 0.426 W/m$^2$.

**Keywords:** blue energy; ion exchange membrane; reverse electrodialysis (RED); salinity gradient energy

## 1. Introduction

There is a growing need to reduce $CO_2$ emissions to the environment due to the Kyoto Protocol and the Intergovernmental Panel on Climate Change (IPCC) Report on carbon dioxide capture and storage [1]. Furthermore, the limited amount of fossil fuels pushes changes in the direction of alternative energy sources [2]. The use of alternative energy sources with reduced or no $CO_2$ emissions can be considered in order to achieve this purpose [1]. Renewable energy as an alternative energy source can be utilized from various sources: power from solar, wind, geothermal, biomass, tidal, ocean, hydro and sea currents [2].

Salinity gradient energy (SGE) also known as "blue energy" is an alternative and maintainable source of energy, which has potential for growth since it was first identified in the 1950s [3]. Salinity gradient power is a new and non-polluting energy source, which can be generated from the difference in the chemical potential between two solutions with different salt concentrations [4], without emission of toxic gases [1,5]. The SGE can be best demonstrated by its reverse process of "desalination". Since the removal of fresh water from seawater requires energy, the reverse of any desalination cycle should release energy in principle [6]. Free energy is generated by combining fresh water with salt water, thereby converting the chemical potential of low salinity water and high salinity water into electrical energy [1,5].

According to the predictions, extractable salinity gradient power in the world is huge. A previous study showed that up to 2 TW salinity gradient energy could be produced assuming the discharge of rivers into the sea, while the release of treated wastewater effluents into the sea could deliver another 18 GW [7]. A very recent study reported that an estimation of 1650 TWh/yr is viable for SGE generation [8]. In another study, 625 TWh/yr of SGE is shown to be possible when river streams are technically utilized [9].

There are three main available technologies to extract SGE: pressure retarded osmosis (PRO), reverse electrodialysis (RED) and capacitive mixing (CAPMIX). PRO benefits from large differences in salinities in feed streams. The dilute solution can be even pure for practical energy generation [10]. Conversely, for RED, the dilute part (e.g., river water compartment) should have a certain level of ionic content to provide electrical conductivity. Highly concentrated brines can be used in PRO, however, in RED, slightly lower concentrations (e.g., seawater for the high saline compartment) are preferred to maintain the physicochemical properties of ion exchange membranes. On the other hand, CAPMIX is comparatively a new technology and is believed to be far from being competitive for RED and PRO [11]. Consequently, RED may be proposed as the most efficient technology to harvest this energy, specifically when seawater and river water are considered as feed.

This technology utilizes stacks consisting of ion exchange membranes that are placed between electrodes in an alternating pattern and separated by spacers to create flow compartments where the feed solutions are transferred [12,13]. When the electrodes are positioned on each side of the membrane stack and the feed solutions having different salt concentrations are circulated through the stack, a generated electrochemical potential difference can be used as a driving force for electricity production. The anion exchange membranes provide the selective transport of anions toward the anode, whereas the cation exchange membranes permit the selective transport of cations towards the cathode. The electrons released at the anode are subsequently transported through an external circuit to the cathode. In the internal circuit of the stack, charge is carried by ions (Figure 1) [1]. With the help of redox reactions at the electrodes, electroneutrality of the solutions flowing through anode and cathode compartments is kept in balance. Therefore, transport of electrons is possible from the anode to cathode through an external circuit. Electrode rinse solution containing a redox couple is circulated in a closed loop through electrode compartments where electrochemical reactions take place. Electrical current created by the electron transport together with the potential difference over the electrodes delivers electrical power [10].

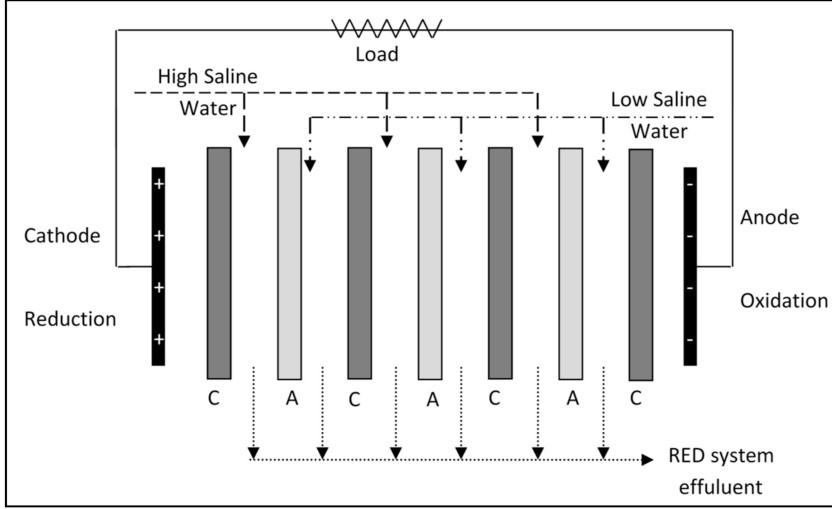

**Figure 1.** Schematic of the reverse electrodialysis (RED) system (C: cation exchange membrane; A: anion exchange membrane).

Similar to other membrane-based systems, the role of ion exchange membranes is of significant importance in the improvement of the RED process. To increase power density and RED process efficiency, several membrane properties such as durability, chemical and mechanical stabilities and structural properties of membranes can be improved. An increasing number of companies are entering the market to produce special ion exchange membranes with enhanced properties and focused on stack and module designs. A number of membrane arrangements comprised of tailor-made and commercial membranes were tested in RED [14]. Hong et al. improved a new and cost-effective method for the fabrication of anion exchange membranes (AEMs) by a hybridization technique [14]. The aim was the development of an ion exchange membrane, particularly in the selection and design of appropriate materials for membranes to be used in RED. The polymers poly (diallyldimethylammonium chloride) and poly (vinyl alcohol) (represented as PDDA and PVA, respectively) were blended in different proportions to form an anion exchange membrane. The highest power density of 0.58 W/m$^2$ was reported by using these membranes [15]. Other studies recently showed that composite pore-filling membranes with relatively thin film thickness improved the RED performance having reduced electrical resistance and high permselectivity [15,16]. Nonconductive polymeric materials and conductive nanopores filled with electrolytes were integrated into the structure of a thin composite pore-filling membrane.

It is recognized that the use of ion exchange membranes made of interpolymers for electrochemical separation could show fruitful results because of their superior electrochemical properties when compared to heterogeneous membranes [17,18]. It is stated that interpolymeric ions exchange membranes, fabricated by extrusion of ethylene/styrene-co-divinylbenzene interpolymer, which was later subjected to chemical modification, could develop the amount of energy produced during mixing solutions with different salt concentrations [18].

In the literature, some publications report different commercial membrane such as Fujifilm membranes for RED tests. Fujifilm membranes have been produced by Fujifilm Manufacturing Europe B.V. (Tilburg, The Netherlands) performed the generation of several types of membrane types with implementing different modifications and improvements for various electromembrane technologies such as electrodialysis (ED), reverse electrodialysis (RED), capacitive deionization (CDI) and electro deionization (EDI) [19]. Vecino et al. studied that a system comprising of a combination of ED and bipolar membrane electrodialysis (BMED) system to achieve the tartaric acid recovery from residues of the winery industry by using homogeneous Fujifilm Type II anion exchange and cation exchange membranes [20]. The performance of the ED system has been evaluated by changing the process parameters such as membrane configuration, membrane types and types of solution used. As a result, the generation of 9.9 g/L of tartaric acid was achieved with a purity of 69.7% $\pm$ 1.3% by using Fujifilm Types II AEM and CEM membranes (Tilburg, Netherlands) [20]. Bhadja et al. performed a comparative study on the EDI system with polyethylene interpolymer based ion-exchange membranes and two commercial membranes as Ionsep and Fujifilm Type II AEM and CEM membranes for the production of ultrapure water [21]. In another study, the transport properties of Fujifilm membranes (AEM Type-I, II, and X and (CEM Type-I, II, and X) have been investigated by Sarapulova et al. and performed a comparative study with homogeneous Neosepta AMX and CMX membranes (Astom, Yamaguchi, Japan) and heterogeneous MK-40 and MA-41 membranes (JSC Shchekinoazot, Tula, Russia) [22]. The results showed that while the transport properties of Type X and Type II membranes are nearly the same as homogeneous Neosepta AMX and CMX membranes, Type I membranes show similar transport properties of heterogeneous MK-40 and MA-41 membranes [22].

Recently, the latest Fujifilm Type 16 produced as the monovalent selective electrodialysis (MSED) membranes have been used by Ahdab et al. for treatment of irrigation water and compared to homogeneous Neosepta MSED membranes [23]. It was observed that the selectivities of Fujifilm MSED membranes were higher than those of Neosepta MSED membranes. When the system is operated with Fujifilm MSED membranes, the cost of the

laboratory scale system is 68% lower than the system with Neosepta membranes [23]. By Rijnaarts et al. the Fujifilm membranes have been used to study the impact of multivalent cations on RED performance and selection of the most suitable cation exchange membranes for the RED system [24]. In that study, heterogeneous Ralex AMH-PES and CMH-PES (MEGA, Stráž pod Ralskem, Czechia), homogeneous monovalent ion-selective Neosepta CMS (Astom Company, Yamaguchi, Japan) and three different homogeneous Fujifilm membranes (FUJIFILM, The Netherlands) like that multivalent ion-permeable Fuji T1 Type I CEM, T0 CEM and Type I AEM have been used. The highest power density (>0.8 W/m$^2$) was obtained by multivalent-permeable Fuji T1 membranes, however, the value of the power density decreased with the addition of Mg$^{2+}$ ions into NaCl solution primarily because of OCV losses [24].

Researchers often worked on laboratory scale RED systems due to the ease of implementation in experiments. Studies operated by large-scale systems were essential for understanding of the feasibility of technologies employed in the real case. For this purpose, four membrane stacks (containing 50 cell pairs) having different sizes from 6 cm × 6 cm, up to 44 cm × 44 cm have been used by Fujifilm membranes provided by Fujifilm Manufacturing Europe BV. Profiled Fujifilm membrane, T1 CEM P150, was used as a cation exchange membrane and standard grade homogeneous Fujifilm type I membrane was used as an anion exchange membrane. The woven net-spacers (Deukum GmbH) of 155 μm were used for separation of all membranes. At the end of the study, it is seen that the efficiency of gross energy obtained by the RED system increases with increasing stack size [25]. The physicochemical properties of all Fujifilm membranes mentioned in this work are summarized in Table 1.

Veerman et al. have found that a RED system contains cell of Neosepta AMX and CMX membranes with 200 μm spacer thickness, with a feed salt ratio of 1:30 (g:g NaCl), obtained 0.65 W/m$^2$ of power density [26]. In another study carried out by Guler et al., power density has reached 1.07 W/m$^2$ for 5 cell pair of Neosepta AMX and CMX membranes with 1:30 of salt ratio [27]. In the same study, for Ralex AMH-PES and CMH-PES membrane pair, power density has been found as 0.60 W/m$^2$.

It is realized from all the previous RED related studies that ion exchange membranes that have homogenous bulk structure have a clear advantage over heterogeneous counterparts. They change less than their electrochemical properties such as electrical resistance and permeability when multivalent ions are encountered and thus RED performance decline with those membranes is less pronounced. Besides, it is easier to make them thinner and (surface) functionalized that increase versatility towards field use. Consequently, it takes great attention to the need for parametric investigation of homogenous membranes and better understands their behavior in RED.

In this study, RED operational parameters such as a number of membrane pairs, salinity gradients and linear flow rates of feed were investigated using a lab-scale RED system equipped with Fujifilm AEM Type 2 and CEM Type 2 ion exchange membranes with homogeneous bulk structure.

**Table 1.** Physicochemical properties of Fujifilm membranes.

| Membrane | Fujifilm Membrane Codes | Membrane Type | Polymer Matrix | Reinforcement | Dry Thickness (μm) | Water Uptake (wt %) | Ion Exchange Capacity (meq/g) | Electrical Resistance (Ω·cm²) | Ref. |
|---|---|---|---|---|---|---|---|---|---|
| Cation Exchange Membranes | Type II CEM | Homogeneous Standard grade | Polyamide | 3D polyolefin fibers structure | 165 ± 5 | 25 ± 5 | 1.35 ± 0.05 | 2.97 | [20–22] |
| | Type I CEM | | | | 120 ± 5 | 29 ± 5 | 1.43 ± 0.05 | - | [22,24] |
| | Type X CEM | | | | 125 ± 5 | 21 ± 5 | 1.67 ± 0.05 | - | [22] |
| | T0 CEM | | | | - | - | - | - | [24] |
| | T1 CEM | Homogeneous multivalent (magnesium) permeable | | | 115 | | - | 1.7 | [24,25] |
| | T1 CEM P150 | Homogeneous Profiled | | | 115/150 [1] | - | - | 2.2 | [25] |
| | Type 10 CEM | | | | 125 | - | - | 2.3 | [25] |
| Anion Exchange Membranes | Type II AEM | Homogeneous Standard grade | Polyamide | 3D polyolefin fibers structure | 165 ± 5 | 10 ± 5 | 1.08 ± 0.05 | 1.55 | [20–22] |
| | Type I AEM | | | | 120 ± 5 | 8 ± 2 | 1.50 ± 0.05 | - | [22,24,25] |
| | Type X AEM | | | | 115 ± 5 | 23 ± 2 | 1.50 ± 0.05 | - | [22] |
| | Type 10 AEM | | | | 125 | - | - | 1.5 | [25] |

[1] Height of profiles on the membrane surface.

## 2. Materials and Methods

### 2.1. RED System

The tests on SGE generation by RED system was carried out by using a lab-scale RED system having 10 cm × 10 cm electrode area. The RED setup includes 2 feed tanks, one with highly saline solution and the other with a low saline solution, an electrode solution tank, peristaltic pumps to feed the solutions into the membrane stack (STT Products B.V. RED stack).

The flow sheet of the RED system is shown in Figure 2. The cross-flow mode was used where the feed streams are connected on one side of the vertically aligned stack. The potentiostat (Gamry Instruments Reference 3000) was employed for monitoring the electrochemical measurements. The RED membrane stack is formed by placing polypropylene mesh (spacer) and silicone gaskets between the sequential cation and anion exchange membranes and placing 2 Ti mesh electrodes (with Ru-Ir coating) at the beginning and end of this sequence. Salinity was measured with a portable conductivity meter (WTW Conductivity meter 3110 model)

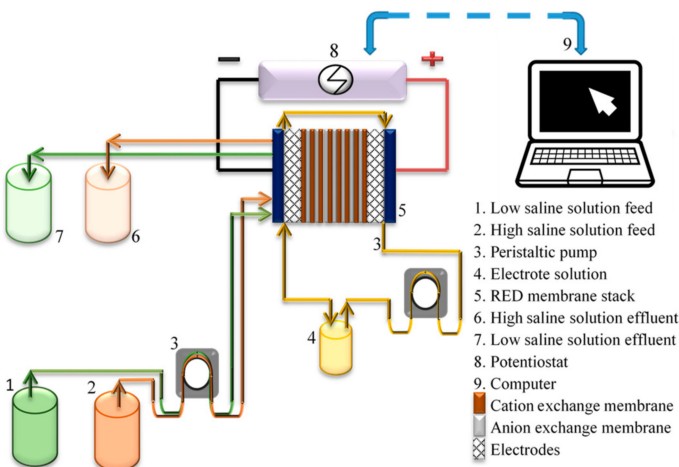

1. Low saline solution feed
2. High saline solution feed
3. Peristaltic pump
4. Electrote solution
5. RED membrane stack
6. High saline solution effluent
7. Low saline solution effluent
8. Potentiostat
9. Computer

▮ Cation exchange membrane
▯ Anion exchange membrane
▨ Electrodes

**Figure 2.** Flowsheet of the lab-scale RED system with 7 membrane pairs (8 CEM + 7 AEM).

### 2.2. RED Tests

For RED tests two different feed solutions, which were low and high saline solutions, were used. The low saline solution (LSS) contained 1 g of NaCl/L, whereas the high saline solution (HSS) 15, 30 and 45 g of NaCl/L representing artificial river water and seawater, respectively, when the effect of salt ratio was investigated. The salt solutions were prepared using analytical grade NaCl. The electrode solution was a mixture of 0.05 M of potassium ferricyanide ($K_3[Fe(CN)_6]$), 0.05 M of potassium ferrocyanide ($K_4[Fe(CN)_6]\cdot3H_2O$) (Pro analyst, Merck, Darmstadt, Germany) and 0.25 M of NaCl. The volumetric flow rate of the solution in electrode compartment was kept constant in all experiments at 300 mL/min. All solutions were prepared using deionized water having a conductivity of 10–15 μS/cm. The technical information about the RED system is shown in Table 2.

**Table 2.** Technical information of the RED system.

| Parameter | Properties | |
|---|---|---|
| Active membrane area (electrode area) | 10 cm $\times$ 10 cm | |
| Electrodes (anode and cathode) | Ti/Ru-Ir alloyed mesh type (mesh 1.0, Area: 10 cm $\times$ 10 cm) | |
| Spacer thickness ($\mu$m) | 400 | |
| Volumetric flowrate of electrode rinse solution | 300 mL/min | |
| Composition of electrode rinse solution | 0.05 M $K_4Fe(CN)_6$/0.05 M $K_3Fe(CN)_6$ and 0.25 M NaCl mixture | |
| Concentrations of feed solutions (20.0 °C) | Low saline 1 g NaCl/L | High saline 15, 30 and 45 g NaCl/L |
| Flow velocities of feed solutions | 0.208; 0.417; 0.625; 0.833 cm/s | |

## 2.3. Parametric RED Studies with Fujifilm Membranes

In this study, effects of the number of membrane pairs, ratio of salinities and feed flow rate on the open circuit voltage, power and power density produced in the RED system were investigated. For these studies, homogeneous Fujifilm AEM Type 2 and CEM Type 2 ion exchange membranes were used for parametric studies. Properties of these membranes were given in Table 3. The number of membrane pairs was changed as 3, 5 and 7 pairs. The ratio of salinity gradient was changed as 1:15, 1:30 and 1:45 (g of LSS:g of HSS). Feed linear flow rates (flow velocities) differed as 0.208, 0.417, 0.625 and 0.833 cm/s. The study of each parameter was performed in three replicates. The given results are based on the average result of three tests.

**Table 3.** Specifications of Fujifilm ion exchange membranes [28].

| Type | AEM Type 2 | CEM Type 2 |
|---|---|---|
| | Homogeneous Anion Exchanger | Homogeneous Cation Exchanger |
| Support material | Polyolefine | |
| Dry thickness ($\mu$m) | 160 | |
| Ion exchange capacity (mmol/g) | $1.08 \pm 0.05$ | $1.35 \pm 0.05$ |
| Electrical resistance ($\Omega \cdot cm^2$) | 5 | 8 |
| Permselectivity (%) | 95 | 96 |
| Burst strength (kg/cm$^2$) | 5.0 | 4.7 |
| pH range | 2–10 | 4–12 |
| Maximum temperature (°C) | 40 | |
| Application areas | Process water purification, wastewater minimization, obtaining tap water from saline water | |

## 2.4. Blank Tests

The RED membrane stack contains n number (i.e., 3, 5 and 7 pairs) of membrane pairs and one additional cation exchange membrane. This cation exchange membrane (CEM Type 2) was added to the membrane stack to prevent leakage of the electrode solution ($K_4Fe(CN)_6$ and $K_3Fe(CN)_6$ salts) through the membranes in the stack, and to allow the circulation at the electrode compartments only. Thus, the effect of the extra membrane on power generation should be excluded to calculate the gross power density of only membrane pairs. This is done by performing the blank tests where only one CEM was in the stack and only the electrode solution was fed. Afterward, the I.V values of the blank tests were subtracted from the ones of the tests including this extra membrane. The blank test was repeated several times before each set of experiments to ensure that the

I.V behavior was maintained at a constant flow rate and concentration of electrode rinse solution.

## 2.5. Calculations

Open circuit voltage and current–voltage analysis were performed by the chronopotantiometric method. The voltage value calculated at the zero current value gives the highest voltage value, in other words, it is called the open circuit voltage (OCV).

The generated electrical power (W) was found by multiplying each current (*I*) with the potential difference (*V*) corresponding to this current (Equation (1)).

$$P = V \cdot I \tag{1}$$

To exclude the effect of extra cation exchange membrane in the stack, the products of $V \cdot I$ determined in the blank tests were subtracted from these power values. Later, the maximum power value was determined and corrected for the total membrane area to calculate the maximum power density. The calculation of maximum power density ($W/m^2$) was carried out using Equation (2) and it is the power divided by the total active membrane area ($m^2$) [29].

$$P_{gross} = \frac{P}{2 \cdot A \cdot N} \tag{2}$$

where $P_{gross}$ is the gross power density ($W/m^2$), $P$ is power (W), $A$ is an effective membrane area of only one membrane ($m^2$) and N is the number of membrane pairs. The voltage value calculated at the zero current value gives the open circuit voltage.

## 3. Results and Discussion

Operational parameters were grouped properly to see the effect of the membrane pair, feed solution, salinity ratio and linear flow rate of the feed solution on power density and open circuit voltage. General results of the study were given in Table 4. Power and maximum power density values were determined after the subtraction of $V \cdot I$ values obtained in blank tests. Error bars were very small; thus they were not included in Table 4 for clarity. In all parametric graphs presented in the next sections, the error bars were included although they were not visible because of their insignificant values.

### 3.1. Effect of Number of the Membrane Pairs on SGE Production

In this group of studies, RED cells were prepared with 3, 5 and 7 membrane pairs of Fujifilm AEM Type 2 and CEM Type 2 membranes to examine the effect of the number of membrane pair on SGE production. The other variables rather than the number of the membrane pairs were the linear flow rate of the feed solutions (0.208, 0.417, 0.625 and 0.833 cm/s) and the salinity ratio of the feed solutions (1:15, 1:30 and 1:45 g NaCl in LSS:g NaCl in HSS). The polarization curves of power density vs. current density data are shown in Figures 3–5 at each salinity ratio.

**Table 4.** Open circuit voltage, power and power density results for all parameters.

| Number of Membrane Pairs | Salt Ratio (g:g) | Volumetric Flow Rate of Feed (mL/min) | Flow Velocity of Feed (cm/s) | Open Circuit Voltage (V) | Power (W) | Maximum Power Density (W/m²) |
|---|---|---|---|---|---|---|
| **3** | 1:15 | 30 | 0.208 | 0.353 | 0.008 | 0.126 |
| | | 60 | 0.417 | 0.356 | 0.008 | 0.126 |
| | | 90 | 0.625 | 0.353 | 0.024 | 0.393 |
| | | 120 | 0.833 | 0.352 | 0.024 | 0.393 |
| | 1:30 | 30 | 0.208 | 0.441 | 0.040 | 0.668 |
| | | 60 | 0.417 | 0.438 | 0.033 | 0.558 |
| | | 90 | 0.625 | 0.437 | 0.017 | 0.285 |
| | | 120 | 0.833 | 0.435 | 0.019 | 0.314 |
| | 1:45 | 30 | 0.208 | 0.489 | 0.025 | 0.410 |
| | | 60 | 0.417 | 0.484 | 0.021 | 0.351 |
| | | 90 | 0.625 | 0.484 | 0.022 | 0.374 |
| | | 120 | 0.833 | 0.484 | 0.029 | 0.488 |
| **5** | 1:15 | 50 | 0.208 | 0.505 | 0.014 | 0.137 |
| | | 100 | 0.417 | 0.596 | 0.014 | 0.138 |
| | | 150 | 0.625 | 0.600 | 0.014 | 0.137 |
| | | 200 | 0.833 | 0.602 | 0.014 | 0.138 |
| | 1:30 | 50 | 0.208 | 0.738 | 0.029 | 0.291 |
| | | 100 | 0.417 | 0.744 | 0.029 | 0.290 |
| | | 150 | 0.625 | 0.747 | 0.030 | 0.303 |
| | | 200 | 0.833 | 0.747 | 0.025 | 0.251 |
| | 1:45 | 50 | 0.208 | 0.800 | 0.038 | 0.382 |
| | | 100 | 0.417 | 0.810 | 0.040 | 0.401 |
| | | 150 | 0.625 | 0.813 | 0.042 | 0.420 |
| | | 200 | 0.833 | 0.814 | 0.043 | 0.426 |
| **7** | 1:15 | 70 | 0.208 | 0.789 | 0.017 | 0.124 |
| | | 140 | 0.417 | 0.825 | 0.022 | 0.158 |
| | | 210 | 0.625 | 0.826 | 0.023 | 0.161 |
| | | 280 | 0.833 | 0.830 | 0.024 | 0.174 |
| | 1:30 | 70 | 0.208 | 1.011 | 0.032 | 0.227 |
| | | 140 | 0.417 | 1.018 | 0.033 | 0.238 |
| | | 210 | 0.625 | 1.023 | 0.036 | 0.254 |
| | | 280 | 0.833 | 1.027 | 0.038 | 0.269 |
| | 1:45 | 70 | 0.208 | 1.106 | 0.049 | 0.352 |
| | | 140 | 0.417 | 1.120 | 0.048 | 0.343 |
| | | 210 | 0.625 | 1.110 | 0.047 | 0.338 |
| | | 280 | 0.833 | 1.146 | 0.049 | 0.349 |

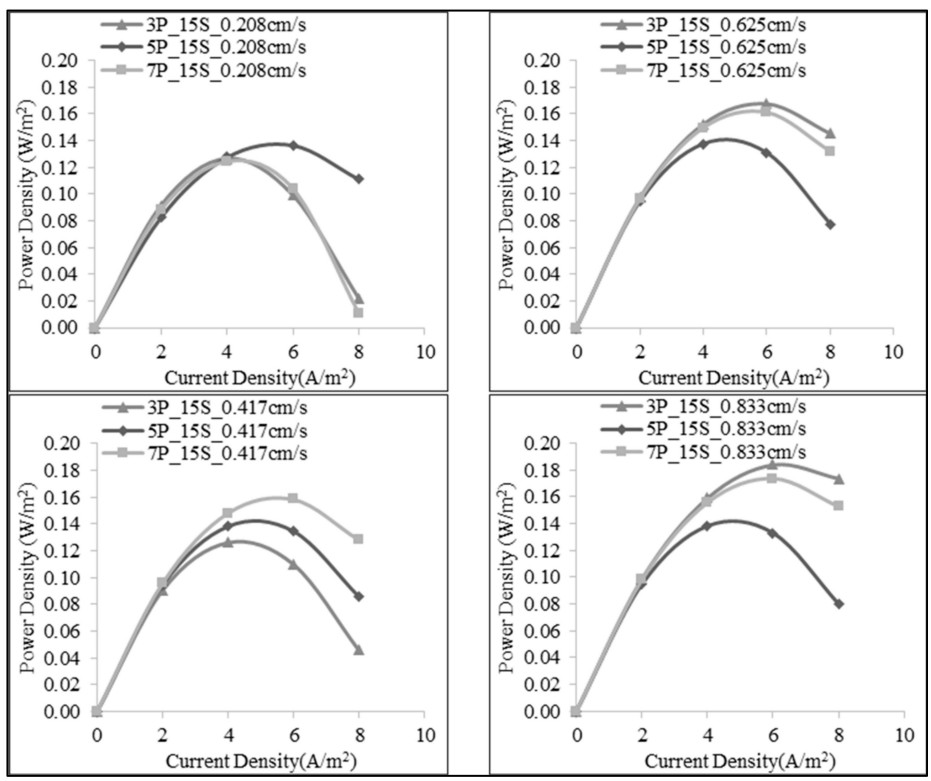

**Figure 3.** Effect of the number of the membrane pair on power density for the salinity ratio of 1:15 (XP_YS_Zcm/s indicates X membrane pair, Y salt content of HSS and Z feed velocity).

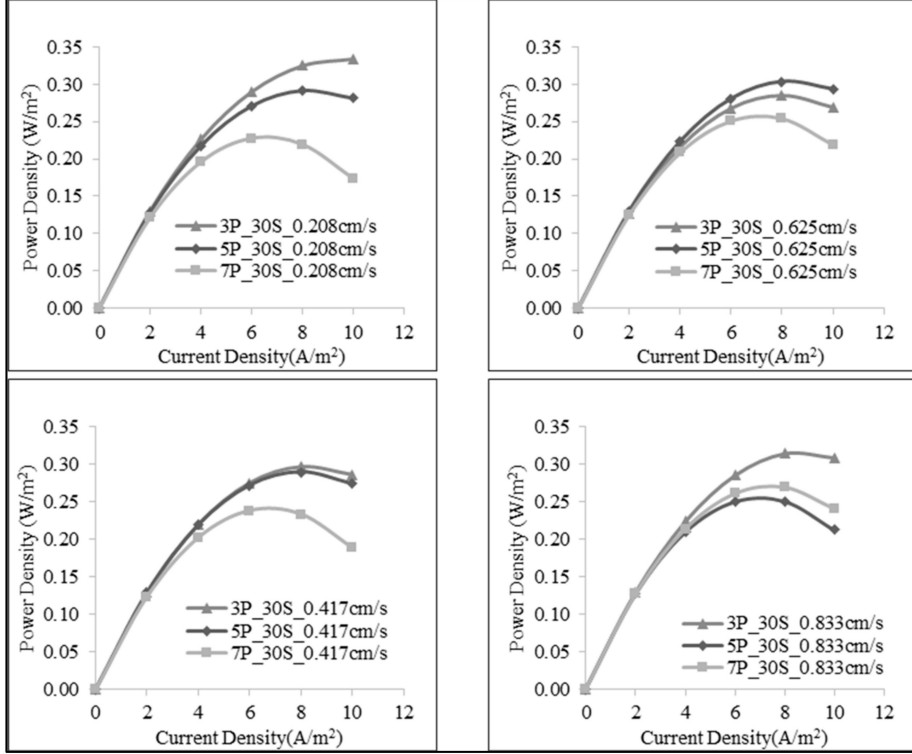

**Figure 4.** Effect of the number of the membrane pair on power density for the 1:30 salinity ratio. (XP_YS_Zcm/s indicates X membrane pair, Y salt content of HSS and Z feed velocity).

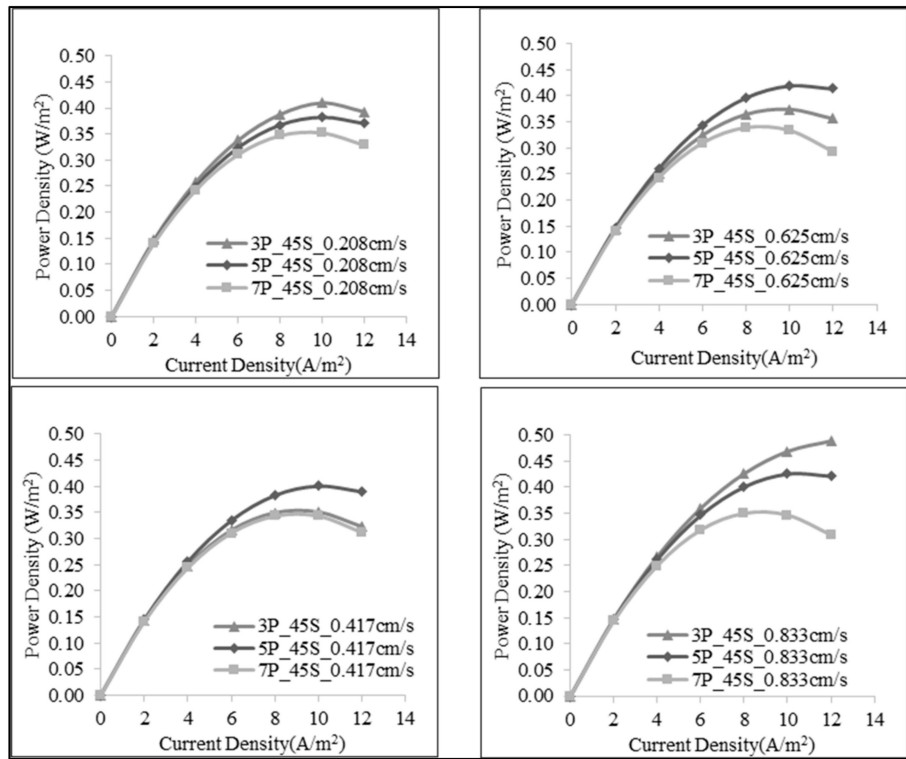

**Figure 5.** Effect of the number of the membrane pair on power density for the 1:45 salinity ratio. (XP_YS_Zcm/s indicates X membrane pair, Y salt content of HSS and Z feed velocity).

This study also shows the effect of upscaling of RED systems by merely varying the membrane pairs keeping the individual size of membranes (10 cm × 10 cm) constant. Previous studies showed the upscaled RED stacks such as 50 cell pairs (with 25 cm × 75 cm and up to 44 cm × 44 cm) [25,26]. Thus, a parametric study was employed in our work in which only the number of membranes (indirectly the number of cell pairs) was varied.

When the power density vs. current density graphs were examined, although a linear relation could not always be observed between 3 and 5 pairs, the highest power densities were mostly achieved in studies with the highest number of membranes, seven pairs at the same experimental conditions (Table 4 and Figures 3–5). This behavior could be due to the internal resistance of the stack. For instance, the proportion of resistance in the electrode compartment was small when a high number of membrane pairs was employed, thus it is not a dominating factor in the generation of power density. Besides, the individual contribution of the resistance in river water (LSS) compartment, which was relatively high in the stack due to the less ionic concentration, was more pronounced in smaller stacks with fewer number of membrane pairs. Consequently, working with smaller membrane pairs might not always show a consistent behavior when parametric work was to be done, thus such investigations should be performed for that specific condition.

In case of the OCV, it is seen that as the number of membrane pairs increased, the open circuit voltage increased also. The highest open circuit voltage was reached with seven membrane pairs and the same trend was observed in all subgroup studies of the effect of number of the membrane pairs (Table 4). This is an expected result as OCV is the sum of voltage over each ion exchange membrane and thus, it is directly proportional to the number of membrane pairs. Although some authors in previous works did not observe any significant reduction in RED performance [25,30], working with a small number of membrane pairs in RED might lead to reduction of power density due to the dominating resistance by some RED components (e.g., electrode resistance and river water resistance) significantly. Consequently, in order to mimic the real behavior of a RED system in the lab,

it is advisable to work with a relatively high number of membrane pairs and overcome the large proportion of resistances inside the stack.

### 3.2. *Effect of Linear Flow Rate of Feed Solutions on SGE Production*

To examine the effect of the feed solutions flow rates, the feed solutions were fed to the RED system at four different velocities such as 0.208, 0.417, 0.625 and 0.833 cm/s. The power density vs. current density graphs were illustrated for seven pairs of membrane in Figure 6. Only when seven pairs of membranes were used, the highest velocity provided the highest power density for each salinity ratio (Figure 6). For the membrane pairs of 3 and 5, it is observed that it was difficult to reach a clear result for the impact of the feed flow rate on power density by Fujifilm AEM Type 2 and CEM Type 2 membranes. The highest maximum power density was observed at different velocities regardless of the membrane pairs and salinity ratio. As there was no clear relation with the number of membrane pairs and salinity ratio, it could be realized that one should carefully analyze the behavior of RED system when there was a case of change in the flow rates for different membrane pairs and salinity ratio. The OCV values in Table 4 for four different linear flow rates were very close to each other, therefore it could be realized that the open circuit voltage did not exhibit a strong relation with the feed flow rate in accordance with the explanation above. In fact, a clear direct relation between the flow rate and OCV was expected such that larger volume flow rates promote slower change in concentrations for both HSS and LSS resulting in a higher cell potential (i.e., OCV) [31]. Nevertheless, optimization of intermembrane thickness is possible that affects the pumping power and thus the net power (i.e., electrical power subtracted by hydrodynamic loss due to transport of fluids) [32].

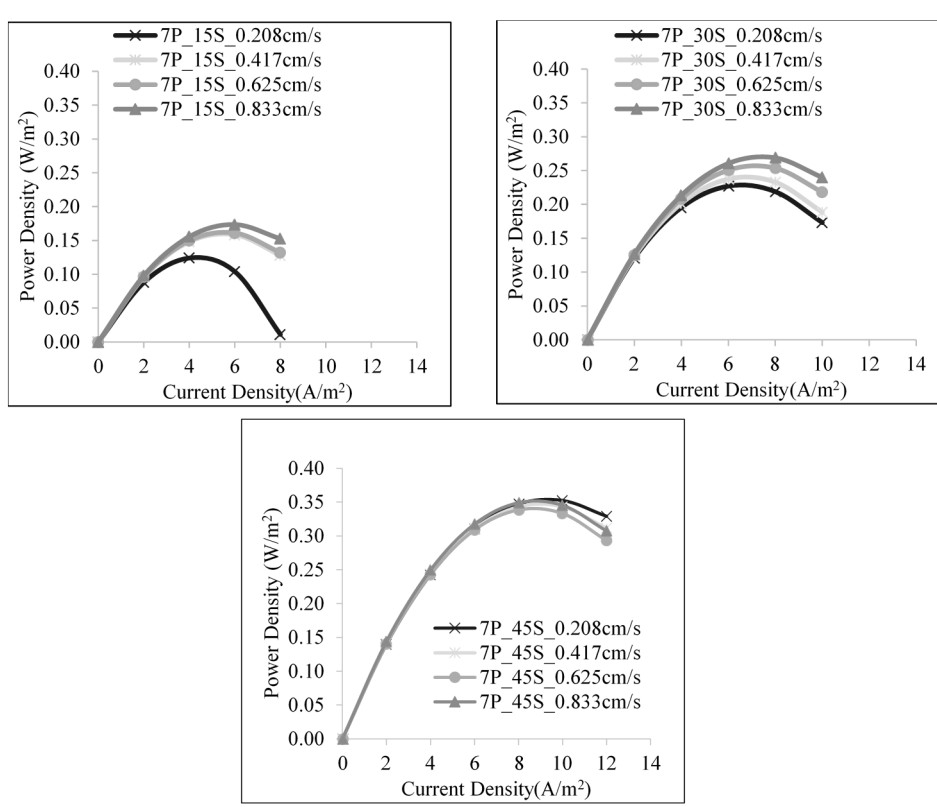

**Figure 6.** Effect of linear flow rate of feed solutions on power density for 7 pairs of membrane. (XP_YS_Zcm/s indicates X membrane pair, Y salt content of HSS and Z feed velocity).

In the case of seven membrane pairs, the behavior of direct proportion is observed between the flow velocity and power density when 1:15 and 1:30 salinity ratios were used because higher feed flow velocities promoted more ionic migration over each ion exchange

membrane. However, the salinity ratio of 1:45 provided closer values of power density but no clear relation between the power density and the flow velocity (Figure 6). High salt content in HSS compartment might exhibit concentration polarization effects and thus caused different power densities at different flow velocities. Another possible cause for this situation is the reduction in membrane permselectivity at highly concentrated salt solutions thus restricting the power generation [33].

### 3.3. Effect of Salt Ratio on SGE Production

To investigate the effect of the salinity ratio of low saline to high saline solution, the low saline solution was kept constant as 1 g/L, while the salinity of the high saline solution was changed to 15, 30 and 45 g/L. These salinity ratios were studied for 3, 5 and 7 membrane pairs and linear flow rates of 0.208, 0.417, 0.625 and 0.833 cm/s. The power density vs. current density plots for each membrane pair were shown in Figures 7–9. In all of the salinity ratio studies examined for each flow velocity and membrane pair, it was found that the power densities increased by the increase in the salinity ratio as also predicted by the Nernst equation in which the potential was directly proportional with the activity (concentration) ratio of salt solutions. In addition, OCV values in Table 4 showed an increase with increasing salinity ratio at the same experimental conditions (i.e., at the same flow velocity using the same number of membrane pairs). Both from the power density and OCV values, it could be deduced that salinity ratio had a strong correlation with the generated power density and had the primary impact in the RED performance.

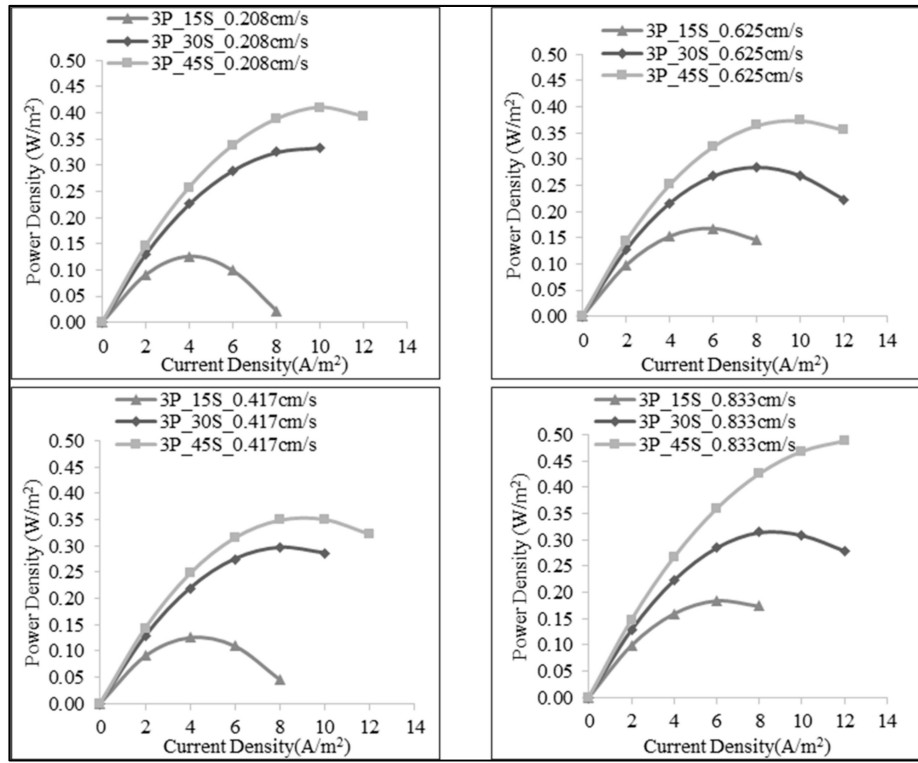

**Figure 7.** The effect of the salt ratio of feed solutions on power density for 3 membrane pairs. (XP_YS_Zcm/s indicates X membrane pair, Y salt content of HSS and Z feed velocity).

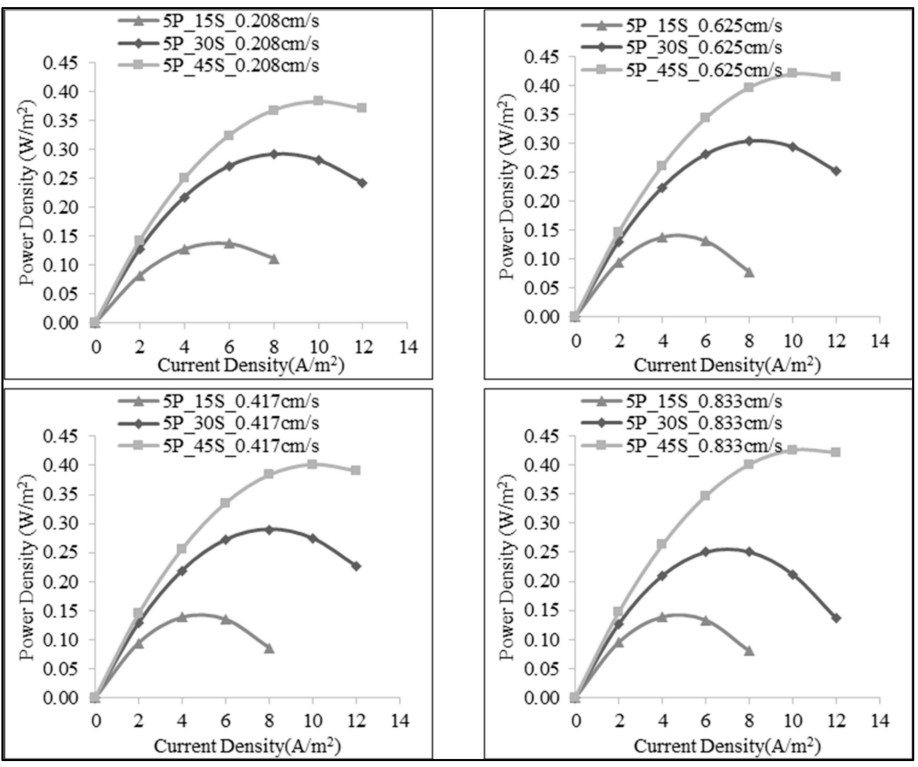

**Figure 8.** The effect of the salt ratio of feed solutions on power density for 5 membrane pairs. (XP_YS_Zcm/s indicates X membrane pair, Y salt content of HSS and Z feed velocity).

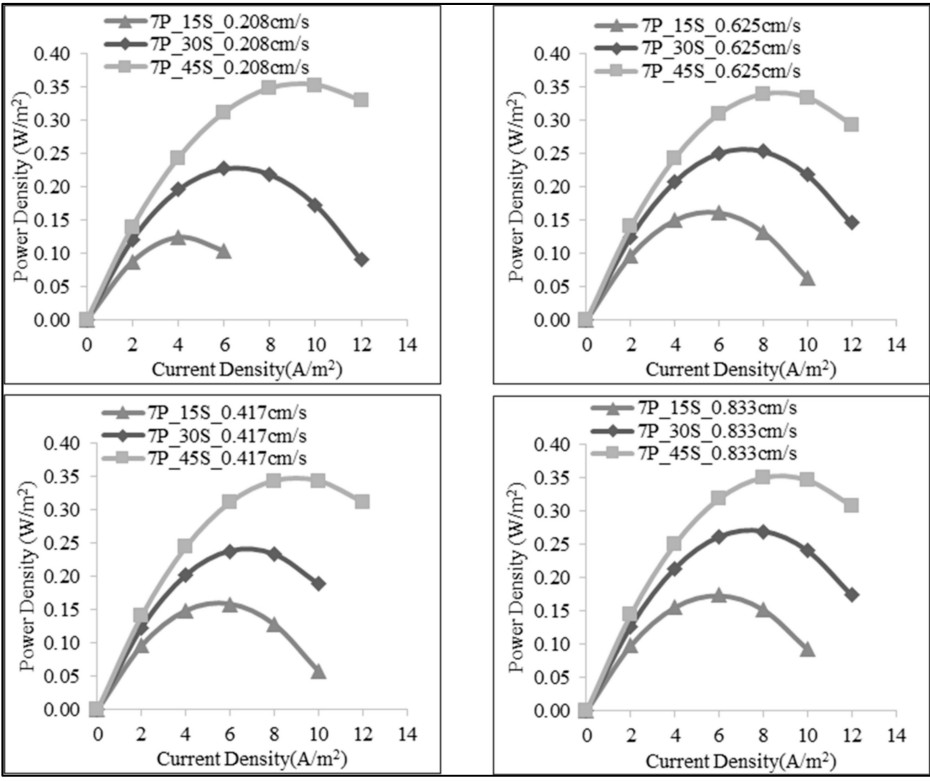

**Figure 9.** The effect of the salt ratio of feed solutions on power density for 7 membrane pairs. (XP_YS_Zcm/s indicates X membrane pair, Y salt content of HSS and Z feed velocity).

It is also worth mentioning that the value of maximum power density obtained in this work was reasonable when compared to the other previous work [29]. At the same operating conditions except the intermembrane distance (i.e., the same number of membranes: 5, salt ratio: 1:30 and flow rate: 0.417 cm/s), Fujifilm membranes exhibited half the power density value ($0.29 \, W/m^2$) of the one obtained by Ralex membranes ($0.60 \, W/m^2$). When higher electrical resistance of Ralex membranes but thinner intermembrane distance is considered, the obtained value of power density becomes valid.

## 4. Conclusions

In these three groups of experiments, the effect of the salt ratio, feed flow rate and number of membrane pairs was examined by using commercially available ion exchange membranes, Fujifilm AEM Type 2 and Fujifilm CEM Type 2 membranes having a homogeneous bulk structure. It was observed that the open circuit voltage and power density increased with the increasing salt ratio, and it had the strongest impact on the RED performance. However, a direct correlation was not observed when the effect of membrane pair and flow velocity were investigated in terms of energy generation. Nevertheless, when the highest number of membrane pair was used, the highest value of power density was achieved. Thus, it was deduced that working with a relatively high number of membrane pairs favored the real RED behavior in an upscaled level. In terms of open circuit voltage, there was a direct proportion with the number of membrane pairs and salinity ratio, but not with the flow velocity. It was seen that the highest power value as 0.049 W was reached with seven membrane pairs and 0.833 cm/s of the linear flow rate by using 1:45 of the salt ratio.

**Author Contributions:** Conceptualization, E.A. and T.Z.K.; Methodology, E.A. and T.Z.K.; Writing—Original Draft Preparation, E.A., T.Z.K. and E.G.; Writing—Review and Editing, E.G., N.K. and M.B.; Supervision, E.G. and N.K.; Project Administration, N.K.; Funding Acquisition, N.K. All authors have read and agreed to the published version of the manuscript.

**Funding:** This research was funded by TÜBİTAK, grant number TÜBİTAK 117M023.

**Institutional Review Board Statement:** Not applicable.

**Informed Consent Statement:** Not applicable.

**Data Availability Statement:** The data presented in this study are available on request from the corresponding author.

**Acknowledgments:** This research has been financially supported by bilateral collaboration program (TUBITAK-NCBR-2549) between Turkey and Poland (Project No: TÜBİTAK 117M023). We acknowledge FUJIFILM Manufacturing Europe BV for sending us Fujifilm AEM Type 2 and Fujifilm CEM Type 2 membranes for our tests. E. Altıok is grateful for the PhD scholarship of Turkish Higher Education Council (YÖK 100-2000).

**Conflicts of Interest:** The authors declare no conflict of interest.

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
