# Peer review of "Performance of Reverse Electrodialysis System for Salinity Gradient Energy Generation by Using a Commercial Ion Exchange Membrane Pair with Homogeneous Bulk Structure"

_water, doi:10.3390/w13060814_

Round 1

Reviewer 1 Report

- Lines 1-4: “Performance of reverse electrodialysis system for production of salinity gradient energy by using homogeneous ion exchange membranes”. Please review the title of the manuscript. The work presented only focuses on one, very specific membrane, thus, it should be clear. Also, it is not possible to produce energy.

- Lines 38-39: “SGE is a new and non-polluting energy source which is the chemical 38 potential energy between two solutions with different salt concentrations [4], without 39 emission of toxic gases [1, 5]” – First of all, I believe you want to say Salinity gradient power and chemical potential difference, but please check this sentence.

- Lines 47-49: “Salinity gradient energy has energy potential of 2.0-2.6 TW, equivalent to 80% of anticipated global electricity production in 2020 or 10% of the total renewable energy, also the total obtainable potential of SGE from natural sources is projected to be about 647 GW [4, 7-9].” Ref. 4 is from 2014 and ref. 7 is from 2012, so what is the point of citing those references, which are forecasting scenarios for 2020?; we are already in 2021. If real data for 2020 are not yet available, or strongly affected by an exceptional decline of industrial activity due to Covid-19, for sure the ones for 2019 should be available.

- Lines 51-54: “There are three main available technologies to extract SGE: pressure retarded osmosis (PRO), reverse electrodialysis (RED) and capacitive mixing (CAPMIX). RED is the most efficient technology to harvest this energy, specifically when seawater and river water are considered as feed.“. Why do you consider that “RED is the most efficient technology to harvest this energy, specifically when seawater and river water are considered as feed”?

- Line 59: “electrical potential can be used as a driving force for electricity production.” A driving force always exists because of some unbalance, so a difference, so it should be an electrochemical potential difference.

- Lines 66-69: “Therefore, electrons can be transported from anode to cathode via electrode rinse solution, which contains a proper redox couple for electrochemical reactions. This electrical current together with the potential difference can be used to produce electrical power [12].” Please, review.

- Lines 100-151 – These lines are almost all about Fujifilm membranes. Could you resume that information in a table? Also, what kind, shape, geometry have the profiles of Fujifilm t1 CEM P150 membrane?

- Please improve Figure 2. Specially the RED stack design. How many cell pairs were used? Was the stack operated in a co-current, a counter-flow or a cross-flow mode?

- Lines 188-189, Table 1, lines 203-204: I was a little bit confused, since in the abstract it is mentioned that the salinity ratio was 1:45. However, first time when the information is given, in lines 188-189 and Table 1 (Technical information of the RED system), the only reported concentrations of feed solutions are 1 and 30 g/ of NaCl per L. Later, in lines 203-204, it is mentioned that it was changed, but that information should also be included in Table 1.

- Lines 215-216 (Blank Tests): “The blank test was repeated before the studies run with 215 each type of membrane.”. I don’t understand how many blank tests were performed. Why more than one? The “extra” membrane is always the same CEM. Only one kind of membrane was used (Type 2 Fujifilm). Or do you mean that blank tests were performed for different flow rates? Sections 2.1-2.4 should be revised, because what should be the same information, it is given in different ways in different subsections.

- Eq.2, if N is the number of membranes, why to multiply it by 2. Or N is the number of membrane pairs? How was the blank resistance discounted?

- In Table 3, are the reported values experimental, or after blank resistance had been discounted? Is the power density gross or net? Actually, what do you mean by net power density in lines 216-219. The net power used commonly in RED is the Gross power – pumping power. You are introducing in lines 216-219 net power as Total power – power in blank test. However, in blank tests, since electrode rinse solution concentration at each side is the same, the OCV should be zero, so how was the power different from zero? Or had you calculated the resistance of the system comprising the extra membrane, rinse electrode solution and the electrodes? Or did you perform 2 tests? One with extra membrane, and the second without any membrane? In lines 212-213 it is written (and that sentence also needs to be rewritten) “The purpose of this calculation is due to the electrical power drop emerged from the electrical resistance of the additional cation exchange membrane” which would suggest that. However, without clear explanation of what was done, I can not recommend this paper for publication, yet. After the authors provide a better description of what was done, and how it was calculated, I will read the rest of the paper with the most interest.

Reviewer 2 Report

This paper deals with the impacts of the salt ratio, feed flow rate and number of membrane pairs on the performance of the RED system using homogeneous ion exchange membranes. The study is interesting, and can be acceptable for publication after following minor revisions. 1. The error bar or uncertainty analysis for the experimental data is needed. As an experimental study, the experiments should be carried out several times to ensure a convincing result. 2. Is the redox potential excluded before obtaining the V-I curves? 3. Recent studies such as the multi-objective optimization for the flow rates “Reverse electrodialysis: Modelling and performance analysis based on multi-objective optimization” and optimization for the channel thickness “ Performance analysis of reverse electrodialysis stacks: Channel geometry and flow rate optimization ” may offer a wider readership. Above recent literatures may help better understanding the impacts of the flow rates on the system performance. 4. In the experiments, the pumping power consumption can be easily obtained. Was the power consumption considered? 5. The power density could be further compared to other existing experimental data.

Round 2

Reviewer 1 Report

N/a